# Viroid Replication, Movement, and the Host Factors Involved

**DOI:** 10.3390/microorganisms12030565

**Published:** 2024-03-12

**Authors:** Yuhong Zhang, Yuxin Nie, Luyou Wang, Jian Wu

**Affiliations:** State Key Laboratory for Managing Biotic and Chemical Threats to the Quality and Safety of Agroproducts, Key Laboratory of Biotechnology in Plant Protection of MARA and Zhejiang Province, Institute of Plant Virology, Ningbo University, Ningbo 315211, China; yuhongzhang0105@139.com (Y.Z.); 15169312489@163.com (Y.N.); wly13858076596@163.com (L.W.)

**Keywords:** viroid, replication, trafficking, host factors

## Abstract

Viroids represent distinctive infectious agents composed solely of short, single-stranded, circular RNA molecules. In contrast to viruses, viroids do not encode for proteins and lack a protective coat protein. Despite their apparent simplicity, viroids have the capacity to induce diseases in plants. Currently, extensive research is being conducted on the replication cycle of viroids within both the *Pospiviroidae* and *Avsunviroidae* families, shedding light on the intricacies of the associated host factors. Utilizing the potato spindle tuber viroid as a model, investigations into the RNA structural motifs involved in viroid trafficking between different cell types have been thorough. Nevertheless, our understanding of the host factors responsible for the intra- and inter-cellular movement of viroids remains highly incomplete. This review consolidates our current knowledge of viroid replication and movement within both families, emphasizing the structural basis required and the identified host factors involved. Additionally, we explore potential host factors that may mediate the intra- and inter-cellular movement of viroids, addressing gaps in our understanding. Moreover, the potential application of viroids and the emergence of novel viroid-like cellular parasites are also discussed.

## 1. Introduction

Viroids, which are unique infectious agents composed exclusively of short, single-stranded, circular RNA molecules, stand out in the realm of microbial entities [1,2]. Diverging from viruses, viroids are naked RNAs, lacking a coat protein. The discovery of viroids in the 1970s represented a pivotal moment in expanding our comprehension of the biosphere [3]. This development came after Antonie van Leeuwenhoek’s 1675 revelation of “subvisible” microorganisms and the discoveries by Dmitri Iosifovich Ivanovsky and Martinus Beijerinck in 1892–1898 about “submicroscopic” viruses. The unique features of viroids led the International Committee on Taxonomy of Viruses to institute a distinct order specifically for these subviral agents [4]. The realm of viroids is predominantly associated with angiosperms, specifically flowering plants, where they manifest as infectious agents causing diseases [5]. The economic significance of these viroid-induced diseases varies widely. Recent advances in metatranscriptomics have unveiled a broader host diversity of viroids and other viroid-like elements than was previously understood [5,6,7,8]. While traditionally linked to plants, the scope now extends beyond the botanical realm, even encompassing prokaryotes [5]. This expanded understanding challenges prior perceptions and opens new avenues for exploring the impact and interactions of viroids in diverse biological systems. As we delve into the intricate world of viroids, their implications are not only confined to the plant kingdom but extend to a more diverse and complex ecological landscape.

The intrigue surrounding viroids has prompted extensive research, particularly in unraveling the intricacies of their replication cycle within the *Pospiviroidae* and *Avsunviroidae* families [9,10,11]. These investigations have not only shed light on replication mechanisms but have also unveiled the indispensable role played by host factors [12]. One prominent model for such investigations is that of the potato spindle tuber viroid (PSTVd, *Pospiviroidae*), chosen for its representation of viroid behavior. Researchers have delved into the intricate world of RNA structural motifs, specifically those involved in facilitating the trafficking of viroids between different cell types. The focus on understanding the molecular intricacies of viroid movement within a cellular milieu has led to comprehensive insights into the replication processes.

Despite the strides made in elucidating viroid replication and trafficking, a critical gap persists in our understanding of the host factors governing intra- and inter-cellular movement. This review aims to consolidate and present our current knowledge of viroid replication and movement within the *Pospiviroidae* and *Avsunviroidae* families. The emphasis will be on the structural prerequisites for these processes and the crucial host factors that have been identified so far. Furthermore, the review delves into uncharted territory, exploring potential host factors that may mediate the movement of viroids within and between cells. By addressing these gaps in our understanding, we strive to contribute to a holistic comprehension of viroid biology and its implications for plant health.

## 2. Viroid Replication and Host Factors Involved

Viroid replication is intricately linked to both their inherent properties and the specific subcellular organelles facilitating this process. A scrutiny of approximately 30 characterized viroids, conducted through sequence and structural analyses, has led to their classification into two families as follows: (i) *Pospiviroidae*, exemplified by the type species potato spindle tuber viroid (PSTVd), with these viroids showcasing a central conserved region (CCR) and primarily adopting a rod-like secondary structure; (ii) *Avsunviroidae*, represented by the type species avocado sunblotch viroid (ASBVd) [13], with members of this family lacking a CCR but exhibiting the capability to form hammerhead ribozymes in both polarity strands. Their secondary structures display either quasi-rod-like or distinctly branched configurations. The categorization of viroid species into genera involves additional criteria, further emphasizing the diversity within this group. Beyond structural disparities, the two families also diverge in their functional properties. Current evidence suggests that while *Pospiviroidae* family members undergo nuclear replication and accumulation, those affiliated with the *Avsunviroidae* family replicate and accumulate within the chloroplast [14,15]. The genomic amplification process within both families is orchestrated through rolling cycle replication (RCR), a mechanism that unfolds differently in each family.

In the *Pospiviroidae* family, genome replication takes place within the host cell nucleus through an asymmetric mechanism (Figure 1). The orchestration of this process involves the host DNA-dependent RNA polymerase II (Pol II), which transcribes the monomeric circular genomic RNA (plus-strand). Extensive research has been conducted on the Pol II-mediated transcription of PSTVd. It has been documented that *N. benthamiana* Pol II interacts with both the (+)- and (-)-strand PSTVd RNAs in vivo. Furthermore, the shorter variant of transcription Factor IIIA (TFIIIA-7ZF) functions as a transcription factor, directly guiding Pol II-catalyzed transcription using the PSTVd RNA genome as a template in plants [16]. Furthermore, PSTVd establishes interactions with ribosomal protein L5 (RPL5), which is a splicing regulator for TFIIIA, both in vitro and in vivo. Infection by PSTVd disrupts the regulatory influence of RPL5 on the splicing of TFIIIA transcripts. Conversely, the overexpression of RPL5 diminishes the expression of TFIIIA-7ZF and hinders the accumulation of PSTVd. This implies that viroids possess the ability to autonomously regulate their replication and impact specific regulatory factors, leading to modifications in the splicing of a restricted set of genes [17]. Furthermore, analysis through nano-liquid chromatography–tandem mass spectrometry utilizing a purified Pol II complex on RNA templates has unveiled modified Pol II lacking Rpb4, Rpb5, Rpb6, Rpb7, and Rpb9 [18]. This contrasts with the conventional 12-subunit Pol II or the 10-subunit Pol II cores observed on DNA templates. The absence of Rpb9, crucial for Pol II fidelity, is noteworthy and provides insight into the heightened mutation rate of viroids in comparison to that of cellular transcripts. This adapted Pol II exhibits transcriptional activity with the assistance of TFIIIA-7ZF and does not seem to require other typical general transcription factors. This observation suggests the presence of a unique mechanism or machinery specifically for viroid RNA-templated transcription [18,19]. Pol II-mediated transcription results in the generation of linear (-)-strand concatemers, acting as replicative intermediates [20,21,22,23]. Here, it is important to distinguish between the (+) strand and (-) strand. The (-)-strand concatemers then serve as a template for Pol II to synthesize linear (+)-strand concatemers. Upon entering the nucleolus [24], these concatemers undergo specific cleavage by RNase III activity, followed by circularization through DNA ligase I, ultimately forming the mature viroid genome [25,26,27]. It is noteworthy that the circular (+)-strand genome monomer predominates as the major form of PSTVd RNA in infected cells. While RNase III is considered the most likely enzyme for cleaving (+)-strand concatemers into monomers, it is essential to note that there is currently no direct in vitro or in vivo evidence demonstrating the cleavage of family *Pospiviroidae* RNA by this enzyme. This conjecture stems from the observation that the (+)-strand monomer RNAs from *A. thaliana*, which transgenically expresses dimeric transcript CEVd (+) RNAs, exhibit characteristics such as 5′-phosphomonoester and free 3′-hydroxyl termini. These features align with the attributes expected of RNase III cleavage products [25]. Moreover, in the study that DNA ligase I was characterized as the enzyme that responsible for the ligation of linear-form (+)-strand PSTVd RNA, DNA ligase 1 was found to specifically and efficiently catalyze the circularization of the linear-form PSTVd transcript, starting from position 96 G and ending with position 95 G and containing the 5′-phosphomonoester and 3′-hydroxyl terminal groups [27]. However, recent studies have shown that multiple RNA transcripts with different starting sites of PSTVd and other viroids in the same family can infect both tomato and *N. benthamiana* plants, while a correct rod-like structure of the RNA transcripts is critical for infection [28,29]. This suggests that the presence of alternative mechanisms or enzymes facilitates the circularization of linear-form PSTVd RNA.

In contrast, the *Avsunviroidae* family adopts a symmetric replication mechanism within the chloroplast (Figure 1). The nuclear-encoded chloroplast RNA polymerase (NEP) takes charge of transcribing the genome to produce (-)-strand concatemers. Using avocado sunblotch viroid (ASBVd) as a model, studies have indicated that purified chloroplast preparations from ASBVd-infected leaves possess the capability to transcribe ASBVd RNAs. Interestingly, tagetitoxin (5–10 mM), known to inhibit the function of the other primary chloroplastic RNA polymerase plastid-encoded polymerase (PEP) but not NEP, does not hinder the transcription of ASBVd strands [30]. This suggests that NEP is the RNA polymerase required for ASBVd replication. However, the potential involvement of another tagetitoxin-resistant RNA polymerase from the chloroplast cannot be ruled out [30]. Moreover, the involvement of transcription factors (TFs) is unknown. Co-transcriptionally, these concatemers undergo cleavage to monomers facilitated by an encoded hammerhead ribozyme, and they are subsequently circularized by a tRNA ligase [30,31,32]. As per reports, a modified version of the chloroplastic isoform of tRNA ligase derived from eggplant (*Solanum melongena* L.) has demonstrated efficient in vitro circularization of both the plus (+)-strand and minus (-)-strand monomeric linear replication intermediates from four viroids within the *Avsunviroidae* family. Furthermore, this RNA ligase exhibits specific recognition for the genuine monomeric linear (+) ELVd replication intermediate, commencing from position 333A and concluding at position 1G. However, it does not display similar recognition for the five other monomeric linear (+) ELVd RNAs with distinct starting positions, despite sharing the same 5′-hydroxyl and 2′,3′-phosphodiester terminal groups [23]. The resulting circular (-)-strand monomers serve as a template for the synthesis of linear (+)-strand concatemers. Being analogous to the *Pospiviroidae* family, these concatemers are cleaved by an embedded hammerhead ribozyme and circularized to generate mature viroid genomes [33].

This intricate dance of molecular events highlights the family-specific nuances in viroid replication strategies. Members of *Pospiviroidae* family opt for a nuclear approach with host factors like Pol II and nucleolar processing, while members of the *Avsunviroidae* family execute replication within the chloroplast using chloroplast RNA polymerase and ribozyme-mediated cleavage. The intricacies of these mechanisms contribute to the diversity and adaptability of viroids, underscoring the need for a comprehensive understanding of their biology. While there is evidence supporting the processing of partial viroid genome RNA from both families in yeast, it is important to note that the replication of whole viroid genome has never been observed for yeast or any other eukaryotic organisms apart from plants [34,35]. However, the replication of viroids in fungi appears to be feasible [6,36], but a critical reassessment is still necessary due to the lack of convincing experiments conducted to analyze viroid infection [37].

## 3. Structural Motifs and Potential Host Factors Mediate the Unidirectional Intercellular Trafficking of Viroids

The observation of directional RNA trafficking highlights the complexity of the gene expression network in multicellular organisms. However, few studies have been carried out to explore the structural and molecular mechanisms that mediate directional RNA trafficking. The potato spindle tuber viroid has been used as a model to study the cellular boundaries for RNA movement and the structural elements required to cross these boundaries. It has been proven that cellular boundaries universally exist between different cell types in the plant leaf. Specifically, cellular boundaries between the epidermal and palisade mesophyll cells, the palisade mesophyll and spongy mesophyll cells, the bundle sheath and mesophyll cells, and the bundle sheath and phloem were characterized. In addition, the structural motifs required for PSTVd to traffic between these cellular boundaries were also identified [38,39,40,41,42,43,44].

Intercellular communication, involving both RNAs and proteins, is crucial for the growth and development of plants. The systemic exchange of RNAs, such as mRNAs, provides novel insights for studying the sophisticated gene expression network [45]. Additionally, viral and subviral agents must move from cell to cell to systematically infect their hosts. Due to the existence of cell walls, plasmodesmata are the only channels for transferring cytoplasmic content between cells. Plasmodesmata, formed by the cytoplasmic membrane, connect adjacent cells to balance cell autonomy and information transfer [46]. However, cell boundaries established by plasmodesmata and the molecular mechanisms that mediate the cell-to-cell and systemic trafficking of RNA are not well studied. In-depth investigations of the systemic gene expression regulation network and the development of novel antiviral approaches require a better understanding of cellular boundaries and an increased comprehension of the molecular basis for RNA to traverse across these boundaries.

To perform functions, RNAs fold into distinct structural motifs. RNA folding typically initiates with canonical Watson–Crick base pairing, resulting in the formation of stems or helices. Subsequently, using the stems as a backbone, internal loops and bulges form. Bases in the loop region are traditionally considered to be unpaired. However, the application of X-ray crystallography, nuclear magnetic resonance (NMR), and cryo-EM in resolving RNA 3D structures has revealed that bases in the “loop” region of the RNA stem-loop structures often enage in non-canonical base pairs, base stacking, and base–backbone interactions [47]. These unique 3D motifs are frequently observed as binding sites for proteins, small ligands, other RNAs, or other regions of the same RNA [47,48]. In comparison to the sequences, the RNA tertiary structure evolves much more slowly [47,48]. Furthermore, the same structural motif often recurs in different RNA molecules, indicating their crucial functions [47]. Characterizing the structural features of RNAs provides novel insights for studying RNA function. Therefore, it is reasonable to hypothesize that RNA structural motifs may also be involved in RNA trafficking.

Infectious viruses and sub-viral pathogens, such as viroids, must traverse cellular boundaries to successfully infect their host plants. Viroids are non-protein-coding RNAs. Therefore, the systemic infection of viroids provides a promising tool to study the molecular basis that supports cell-to-cell and systemic RNA trafficking [49,50]. We have employed the potato spindle tuber viroid (PSTVd) to investigate the roles of unique structural motifs in mediating RNA trafficking.

The 359 nt circular RNA genome of PSTVd folds into a secondary structure composed of 27 loops flanked by 26 stems (Figure 2A), formed by canonical base pairs [51]. Although no canonical base pairs are formed in loops, they are functional units crucial for the replication and/or trafficking of PSTVd [38]. Notably, loop 27, which is structurally similar to the loop region of the histone 3′ UTR mRNA stem-loop in animal cells, mediates the unidirectional trafficking of PSTVd from epidermal to palisade mesophyll cells—an early step of systemic infection in *N. benthamiana* (Figure 2B) [39]. The structural model of loop 27 contains no base pairing except for the closing cis Watson–Crick (cWW) base pair, suggesting that base stacking is likely to be the key interaction for its function (Figure 2B). The movement protein of the tobacco mosaic virus was identified as being capable of substituting the function of PSTVd loop 27 in the directional movement from the epidermal cells to the mesophyll. This serves as a notable illustration of the functional substitution of RNAs by proteins [40]. Further studies could investigate the potential for the functional substitution of viral proteins by PSTVd RNA.

The movement of PSTVd from the palisade mesophyll to the spongy mesophyll is facilitated by loop 6, which has a 3D structure formed by three non-Watson–Crick base pairs (Figure 2B) [41]. However, whether the trafficking of PTSVd from the palisade mesophyll to the spongy mesophyll is unidirectional remains an open question. The structural motifs mediating the unidirectional trafficking of PSTVd between rgw mesophyll cells and bundle sheath, including the classical pathogenicity and right-terminal domain, have been identified through the functional mutagenesis of two PSTVd strains named PSTVd^NT^ and PSTVd^NB^. The bipartite motif present in the PSTVd^NT^ strain but not in the PSTVd^NB^ strain is not only required but also sufficient for the movement of PSTVd from the mesophyll to the bundle sheath in young tobacco (*Nicotiana tabacum*) leaves [42].

In two of our previous studies, the 3D structure of loop 7, formed by a cWW base pair with water insertion as well as two G/U base pairs on positions 76:283 and 156:205, determines the unidirectional trafficking of PTSVd from the bundle sheath to the phloem (Figure 2B) [43,44]. Our studies suggest that PSTVd may undergo global conformational alterations to enable unidirectional trafficking between the mesophyll cells and the bundle sheath, as well as between the bundle sheath and the phloem. All the cellular boundaries identified in our previous studies are shown in the schematic diagram of a transverse section of a young *N. benthamiana* leaf (Figure 2B). Additionally, structural motifs that mediate the trafficking of PSTVd across the boundaries are indicated on the PSTVd secondary structure (Figure 2A).

In summary, RNA 3D motifs participate in all the steps of the cell-to-cell and systemic trafficking of PSTVd and possibly all RNAs too. The evidence further suggests that directional plasmodesmal gates may differ between all cell types, allowing for the precise regulation of RNA transportation and the establishment of distinct cellular boundaries. The intricate RNA trafficking system established by plasmodesmata also contributes to shaping the sequence structure of PSTVd quasispecies [52].

As we mentioned above, viroids encode no proteins; consequently, all their functions rely on interactions with host factors through their sequences and structures. However, the host factors involved in the trafficking of viroids across specific cellular boundaries remain poorly understood. Viroids exhibit intercellular movement through two main mechanisms as follows: cell-to-cell movement through plasmodesmata and long-distance trafficking facilitated by the phloem. Currently, there is no direct evidence demonstrating that viroid trafficking through the phloem is reliant on host factors. The *Cucurbita maxima* phloem protein 2 (CmPP2) is acknowledged for its capacity to expand the size exclusion limits of plasmodesmata, facilitating its migration from the companion cells where they originate to sieve tubes [53,54]. Additionally, there is evidence of CmPP2 translocating in intergeneric grafts, suggesting its movement within the assimilated stream towards sink tissues [55,56]. While there is no documented direct interaction between CmPP2 and viroids, several studies have explored the connection between viroids and its homolog in cucumber [57,58]. The cucumber phloem protein 2 (CsPP2) has been observed to interact with the hop stunt viroid (HSVd, *Pospiviroidae*) in vivo. Interspecies grafting experiments have indicated that both the CsPP2 and HSVd are translocated to the scion [57]. Additionally, in vivo interactions between the CsPP2 and the apple scar skin viroid (ASSVd, *Pospiviroidae*) have been established [58]. One can infer that the PP2 has the capability to bind specific viroids, thereby contributing to the phloem translocation of these viroids. Both viroids fall under the *Pospiviroidae* family; however, there is no reported information regarding the interaction between *Avsunviroidae* viroids and potential phloem trafficking factors.

For cell-to-cell movement, it is postulated that PSTVd loop 27 may interact with a host factor similarly to how the histone 3’ UTR stem-loop structure binds to the stem-loop-binding protein [39]. This implies the existence of a trafficking protein capable of mediating the unidirectional movement of the PSTVd from the epidermal to palisade mesophyll. An analysis of the model structure of loop 6 suggests that this loop likely serves as a docking site for a Lys residue of a host protein, thereby directing the movement of the PSTVd from the palisade mesophyll to the spongy mesophyll [41]. However, up to now, no host factors that bind to the PSTVd to mediate its movement from cell to cell have been characterized. Nevertheless, knowledge about the host factors orchestrating the intricate cellular trafficking of the PSTVd remains incomplete. Future studies are essential to identify and characterize the functions of these host factors, shedding light on the intricate mechanisms that facilitate the directional movement of the PSTVd between different cell types. This ongoing research not only contributes to our understanding of viroid biology but also holds broader implications for deciphering the complexities of RNA trafficking in plant cells.

## 4. Intracellular Movement of Viroids and Host Factors

After entering the cell, viroids need to undergo intracellular movement to reach the nucleus or chloroplast for replication and leave these organelles after replication. After entering plasmodesmata, the journey of viroids through the cytoplasm to reach their replication sites marks the initial phase of their intracellular movement (Figure 3). By employing microinjection techniques with a fluorescein-labeled PSTVd, it has been disclosed that it takes approximately 23 min to achieve the half-maximal nuclear accumulation of the viroid. This implies the presence of specific host machinery guiding or directly facilitating the movement of the PSTVd through the cytoplasm [59]. The cytoskeleton, a dynamic network of protein filaments, imparts structural support to cells and participates in various cellular processes, including that of intracellular transport [60,61]. Myosins, motor proteins responsible for movement along actin filaments, have long been considered key players in the movement of certain viral proteins in plant cells [62,63,64]. However, studies have indicated that disrupting microtubules and actin filaments with substances like oryzalin or cytochalasin D did not hinder the import of the fluorescein-labeled PSTVd. This suggests that the intracytoplasmic movement of the PSTVd may not be exclusively dependent on the cytoskeleton [65]. However, there is still a need for comprehensive investigations into the role of myosins in the intracellular movement of the PSTVd and other viroids. This can be achieved through the application of diverse methodologies, including the inhibition of gene function through RNA silencing, genomic modifications, and other relevant approaches. Supporting this argument, a recent study demonstrated that, in contrast to the previous notion of chloroplast clustering around the nucleus as a viral defense strategy [66], chloroplast clustering unexpectedly facilitates infection of the PSTVd [67], very likely by promoting the intracellular movement and nuclear import of the PSTVd. Although the role of myosins in chloroplast movement is unclear, certain myosin proteins have been shown to localize at the stromule of chloroplasts (plastid stroma-filled tubules facilitating the signal exchange between organelles as well as between organelles and the cell nucleus) and to participate in stromule movement [68]. After replication, viroids may hijack the same cellular machinery, utilizing it for their movement from the nucleus to plasmodesmata.

The administration of GTP and GDP analogs, specifically GTP (γS) and GDP (βS), did not show any impact on the nuclear import of the PSTVd. This implies that the Ran–GTPase cycle, which typically facilitates the nuclear transport of numerous proteins and nucleic acids, is unlikely to play a role in the nuclear import of the PSTVd [65]. A recent study has indicated that the PSTVd may enter the cell nucleus through the nuclear import pathway by binding to viroid RNA-binding protein 1 (VIRP1) in conjunction with IMPORTIN ALPHA-4 (IMPa-4) [69]. This indicates that the PSTVd may potentially leverage the host’s inherent transport pathways for its intracellular movement. Additionally, VIRP1 exhibits a specific binding affinity for the right-terminal C-loop structure of the PSTVd [70,71], suggesting the involvement of RNA structural motifs in the intracellular movement of the PSTVd. Nevertheless, this conclusion awaits further confirmation through more direct experimental approaches, such as microinjection experiments demonstrating the impaired nuclear import of the PSTVd following the silencing of IMPa-4. Following the entry of the PSTVd into the nucleus, its circular (+) RNA undergoes an initial transcription into the nucleoplasm by RNA polymerase II, resulting in the formation of a linear (-) PSTVd concatemer. Subsequently, this linear (-) PSTVd concatemer acts as a substrate for RNA polymerase II, leading to the production of linear concatemeric (+) RNA. Upon import into the nucleolus, the concatemeric (+) strand undergoes cleavage into unit-length monomers by RNase III [24]. Further investigation is needed to understand the movement of the PSTVd from the nucleoplasm into the nucleolus.

As of now, the mechanisms governing the nuclear export of viroids, including the specific RNA structural motifs and host factors involved, remain unknown. However, insights can be drawn from the export pathways employed by other RNA molecules. For instance, mRNAs utilize the dimeric receptor TAP/p15 for export [72], while miRNAs rely on HASTY, an ortholog of mammalian EXPORTIN5 (XPO5). These analogous pathways suggest that viroids may employ similar cellular mechanisms for their nuclear export, although a direct confirmation is required [73].

The intracellular movement of *Avsunviroidae* viroids has not been thoroughly investigated. Nevertheless, no host RNAs have been identified with the same capability as these viroids in traversing the bilayer plastid membrane and entering the chloroplast. Despite this limited study, Avsunviroidae viroids hold potential as tools for delivering foreign RNAs into chloroplasts [74,75], making them appealing for researchers engaged in plastid genome editing and genetic transformation [76,77]. Given the crucial role of motor proteins in facilitating the intracellular movement of viruses [62,63,64], their potential involvement in the intracellular transport of *Avsunviroidae* viroids is worthy of further investigation.

## 5. The Emergence of Viroid-like Cellular Parasites Requires in-Depth Investigation

Recent progress in metatranscriptomics has expanded our understanding of host diversity for viroids and viroid-like elements. Previously associated mainly with plants, this scope now extends beyond the botanical realm, even encompassing prokaryotes [5,7]. In a recent study, over 5000 metatranscriptomes and 1000 plant transcriptomes underwent analysis. This examination revealed 11,378 circular and covalently closed RNAs with viroid-like characteristics distributed across 4409 species-level clusters. This represents a five-fold increase compared to the previously identified viroid-like elements [5]. More recently, the identification of “Obelisks” signifies an unrecognized class of viroid-like elements initially discovered in the metatranscriptomic data of human gut microbiomes [7]. The identification of these viroid-like RNAs is predominantly based on computer-based predictions of their circular nature and the characteristic secondary structure found in viroids. However, before labeling them as viroids or viroid-like cellular parasites, it is crucial to test their ability to infect hosts and analyze their life cycles. This approach mirrors investigations conducted on fungal ambiviruses, which are viroid-like elements undergoing rolling circle replication [8].

## 6. Future Perspectives

After the identification of microorganisms in 1675 and the discovery of viruses between 1892 and 1898, the unveiling of viroids in 1971 marked the third significant expansion of our understanding of the biosphere [3,78,79]. Over the past half-century, substantial progress has been made in the field of viroid research. This includes advancements in understanding genome sequences, secondary structures critical for successful infection, three-dimensional motifs governing their unidirectional trafficking, ribozyme catalyzing self-splicing, host ranges, transmission modes, quasispecies evolution, anti-viroid approaches, and the host factors involved in both replication and movement [20,52,80,81,82]. However, further characterization of the host factors interacting with viroids is essential to gain fundamental knowledge. Additionally, there is a growing interest in exploring viroids and viroid-like RNAs in animal cells.

## Figures and Tables

**Figure 1 microorganisms-12-00565-f001:**
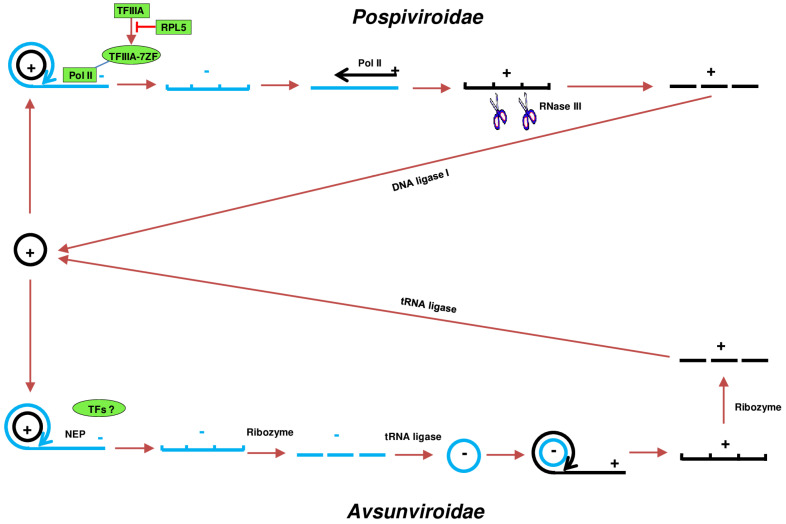
Replication cycles of viroids in two families. *Pospiviroidae* viroids undergo replication in the nucleus. The circular (+) RNA is initially transcribed in the nucleoplasm by RNA polymerase II, resulting in the generation of a linear (-) PSTVd concatemer. This linear (-) PSTVd concatemer is then utilized as a substrate for RNA polymerase II to produce linear concatemeric (+) RNA. After being imported into the nucleolus, the concatemeric (+) strand undergoes cleavage into unit-length monomers by RNase III. Finally, these unit-length monomers are circularized by DNA ligase I. In contrast, *Avsunviroidae* viroids replicate in the chloroplast. The circular (+) RNA is transcribed initially by nuclear-encoded chloroplast RNA polymerase (NEP) to synthesize a linear (-)-strand concatemer, followed by cleavage into monomeric form mediated by hammerhead ribozymes. Subsequently, the (-)-strand monomer serves as a template for NEP to synthesize a linear concatemeric (+) RNA, followed by the second cleavage to form monomers, which is also mediated by hammerhead ribozymes. The linear monomeric (+) RNA is then circularized by intramolecular ligation by tRNA ligase.

**Figure 2 microorganisms-12-00565-f002:**
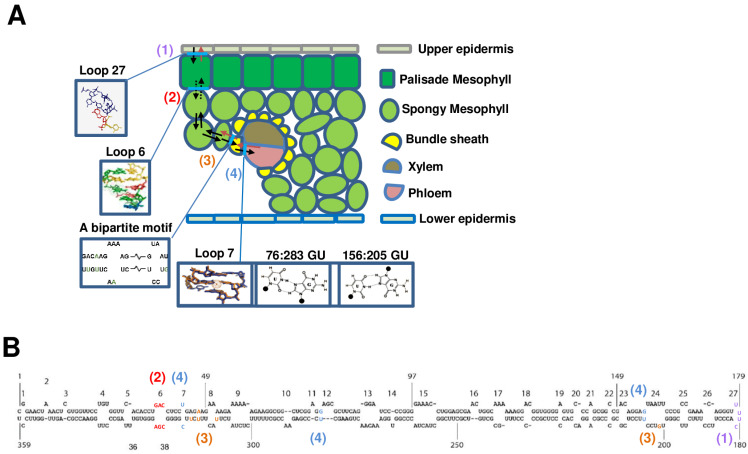
The boundaries between different cell types in plant leaves and the structural motifs necessary for PSTVd to traverse these boundaries. (**A**) The 359 nt secondary structure of PSTVd is shown. (**B**) Illustration of boundaries between different cell types in plant leaves and PSTVd structural motifs required to cross these boundaries. Light blue lines indicate PSTVd structures required to cross specific boundaries between (1) the epidermal and palisade mesophyll cells, (2) the palisade mesophyll and spongy mesophyll cells, (3) the bundle sheath and mesophyll cells, and (4) the bundle sheath and phloem. Structures include 3D loop structures and G-U base pairs. Structural motifs required to cross specific cellular boundaries are labeled on (**A**), with numbers and colors corresponding to cellular boundaries.

**Figure 3 microorganisms-12-00565-f003:**
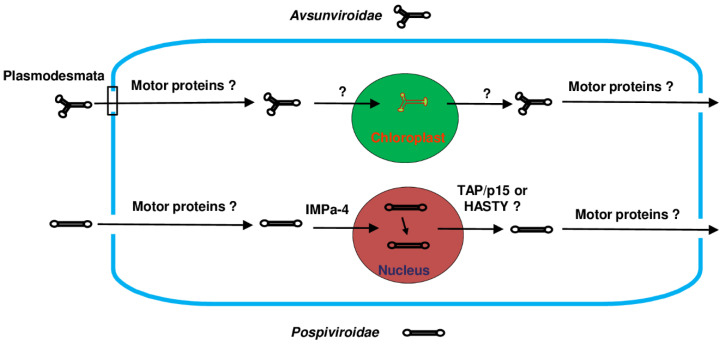
Models depicting the intracellular movement of viroids within two families. Upon entering the cell, viroids exhibit movement towards the nucleus (family *Pospiviroidae*) or the chloroplast (family *Avsunviroidae*), where they undergo replication. Viroid trafficking pathways are indicated by arrows, with characterized and potential host proteins involved in the intracellular movement of viroids also being highlighted. As of now, the host factor identified for mediating the nuclear import of the PSTVd is IMPORTIN ALPHA-4 (IMPa-4). Given their pivotal roles in facilitating the intracellular movement of viral RNA, motor proteins are considered to be potential host factors for mediating the intracellular movement of *Pospiviroidae* viroids. Export pathways for mRNAs involve the dimeric receptor TAP/p15, while miRNAs rely on HASTY, an ortholog of mammalian EXPORTIN5 (XPO5). It is conceivable that TAP/p15 or HASTY may also play a role in mediating the nuclear export of *Pospiviroidae* viroids. In the case of *Avsunviroidae* viroids, motor proteins may be utilized for movement through the cytoplasm. However, speculation on host factors involved in chloroplast import and export is challenging as no host RNAs have been reported to possess the ability to enter and exit the chloroplast.

## Data Availability

The data presented in this study are available on request from the corresponding author.

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
