# Peer review of "Viroid Replication, Movement, and the Host Factors Involved"

_microorganisms, 2024, doi:10.3390/microorganisms12030565_

Round 1

Reviewer 1 Report

Comments and Suggestions for Authors

Zhang and collaborators present a review covering different aspects of the replication and movement (intra- and inter-cellular) of pospiviroids and avsunviroids and the identified host factors involved. The manuscript is well written and can be accepted for publication after minor revision.

Minor issues:

Line 8: include circular RNA

Line 9: Suggestion: In contrast to viruses, viroids do not encode for proteins and lack a protective coat protein.

Line 25: include circular RNA

Lines 26-27: Suggestion: Diverging from viruses, viroids are naked RNA, lacking a coat protein.

Line 64: change characteristics to properties

Lines 117-119: Improve because this sentence does not make sense.

Line 119: amend: catalyze

Lines 117-126: These sentences are repeated.

Lines 170-171: Suggestion: Members of Pospiviroidae opt for... while members of Avsunviroidae execute...

LIne 186: change pass to cross

Line 195: change complicated to sophisticated

Lines 221-222: Suggestion: Viroids are non-protein-coding RNA.

Line 234: tobacco mosaic virus

Line 247: tobacco (Nicotiana tabacum).

Reviewer 2 Report

Comments and Suggestions for Authors

This review provides a detailed summary of the present understanding of how viroids replicate and move. It highlights the structural aspects and host factors involved in this process. The article discusses the structural requirements that viroids need for replication and movement and provides insights into the complex interactions that occur between viroids and their host factors. Overall, this is a well-written and informative review. There may be a few minor points that could be improved to enhance the quality of the manuscript.

Figure 1. Please show the interplay between TFIIIA-7ZF and RPL5 during RNA transcription of PSTVd in Figure 1. Additionally, please do a comparison if there is the presence of transcription factors in the ASBVd family and highlight them in Figure 1.

Figure 2B. Please rearrange the figures and place Figure 2B before Figure 2A.

Line 227. Please add reference “Figure 2B”

Line 232-233. Please add reference “Figure 2B”

Line 96, Please provide a brief explanation of "modified Pol II"? Also, please explain why and how it is modified, is it because of PSTVd infection?

Line 117-125. The sentence was repeated twice.

Line 174-176. The sentence is confusing. Please rewrite it.

Line 180. Please discuss the structural motifs required for ASBVd to traffic between cellular boundaries, if any research has been done.

The following sentences lack proper references. Please provide relevant sources to support them:

1: Line 96. “…utilizing a purified Pol II complex on RNA 95 templates, has unveiled a modified Pol II lacking Rpb4, Rpb5, Rpb6, Rpb7, and Rpb9”

2: Line 152. “…did not hinder the transcription of ASBVd strands. ”

3: Line 191. “…structural 190 motifs required for PSTVd to traffic between these cellular boundaries were also identified. ”

4: Line 209-212. “…has revealed that bases in the "loop" region of RNA stem-loop 211 structures often engage in non-canonical base pairs, base stacking, and base-backbone interactions”

5: Line 214-215. “In comparison to 214 sequences, RNA tertiary structure evolves much more slowly. ”

Line 104. Make the word consistency, “minus-strand “ and Line 105 (-)-strand

Line 278-279. “cell-to-cell movement through plasmodesmata and long-distance trafficking 278 facilitated by phloem. ”
